# Assessment of Immune Responses to Rabies Vaccination in Free-Ranging Dogs in Bengaluru, India

**DOI:** 10.3390/vaccines11050888

**Published:** 2023-04-24

**Authors:** Vinay Chavan Prakash Rao, Sharada Ramakrishnaiah, Shrikrishna Isloor, Rathnamma Doddamane, Dilip Lakshman, Manjunath Shinde Sundar Rao Maralavadi, Avinash Bhat, Balaji Chandrashekar, Krithiga Natesan, Ganesh Kondabattula, Nagendra R. Hegde

**Affiliations:** 1KVAFSU-CVA Rabies Diagnostic Laboratory, WOAH Reference Laboratory for Rabies, Department of Veterinary Microbiology, Veterinary College, KarnatakaVeterinary, Animal and Fisheries Sciences University (KVAFSU), Bengaluru 560024, India; 2Bruhat Bengaluru Mahanagara Palike (BBMP), Bengaluru 560002, India; 3Trouw Nutrition India Pvt. Ltd., Hyderabad 500032, India; 4Project Lead, Mission Rabies, Bengaluru 560072, India; 5ICAR-Indian Veterinary Research Institute, Bengaluru 560024, India; 6National Institute of Animal Biotechnology, Hyderabad 500032, India

**Keywords:** rabies, vaccination, immune response

## Abstract

Rabies is a fatal encephalomyelitis mainly transmitted to humans and other animals by rabid dog bites. Hence, vaccination programs are being instituted for the control of rabies in dogs. Though stray dogs have been vaccinated for years under various programs initiated for control of the disease, the effectiveness of these programs can be ascertained only by assessing the immunity of these dogs. With this in view, a study was conducted to assess the effectiveness of the ongoing mass dog vaccination (MDV) program by the Bengaluru City Municipal Corporation, Bengaluru, India. Whole blood and serum samples (n = 260) from vaccinated stray dogs in 26 wards of 8 corporation zones were tested by rapid fluorescent focus inhibition test (RFFIT) as well as an in-house quantitative indirect enzyme-linked immunosorbent assay (iELISA) for a humoral response and by interferon-gamma (IFN–γ) ELISA for a cellular response. As determined by the cut-off value of 0.5 IU/mL of serum, 71% and 87% of the samples from vaccinated dogs revealed adequate levels of antibodies presumed to confer protection by RFFIT and iELISA, respectively. The sensitivity and specificity of the iELISA were 100% and 63.3%, respectively. The IFN–γ ELISA revealed adequate cellular response in 50% of the samples. The quantitative iELISA was found to be useful in large-scale seromonitoring of MDV programs to aid in the elimination of dog-mediated rabies.

## 1. Introduction

Rabies is an acute, viral encephalomyelitis caused by a *Lyssavirus* belonging to the family *Rhabdoviridae* [1]. It is a disease of the central nervous system (CNS), which usually spreads when an infected animal bites a human or another animal. The virus spreads from the saliva of infected animals when it comes in contact with the eyes, mouth, or nose of the bitten animal or human. Progressive encephalitis begins days or months after infection and ends in the death of the infected individual or animal, typically within 30 days of the onset of symptoms [2]. The case fatality rate (CFR) is nearly 100%, which is the highest among known infectious diseases. There is no specific treatment for rabies; however, it can be prevented by proper vaccination.

Dogs are the most important reservoirs of rabies in Asia. The World Health Organization (WHO) has estimated that 59,000 humans die annually due to dog-mediated rabies, the majority occurring in Asia (59.6%) and Africa (36.4%) [3,4]. The prevalence of rabies is particularly high in India, with as many as 20,000 human deaths every year [5]. The primary source of human infection (in more than 96% of cases) is the unvaccinated free-ranging stray dog population [4,6]. Considering the seriousness of the disease and its endemicity, especially in African and Asian continents, the Tripartite Alliance, which includes the Food and Agriculture Organization (FAO), the World Organization for Animal Health (WOAH), and the WHO along with the Global Alliance for Rabies Control (GARC), have launched “Zero by 30: The Global strategic plan to end human deaths from dog-mediated rabies by 2030” [7].

The elimination of rabies in humans depends on the elimination of rabies in dogs. This can be achieved through postexposure prophylaxis (PEP) of exposed patients, pre-exposure prophylaxis (PrEP) of people at high risk, control of infection in animal reservoirs, and management of a free-ranging dog population [3]. The mass dog vaccination (MDV) campaigns are expected to aid in improving the herd immunity levels of free-ranging dogs and prevent potential human exposure to the rabies virus.

In view of the severity of the disease in India and the need to control the disease, the National Centre for Disease Control (NCDC), Government of India (GOI) launched The National Action Plan for Rabies Elimination (NAPRE) on 28 September 2021. This action plan focuses on strategies to prevent and control rabies to achieve the elimination of dog-mediated rabies by the year 2030. Before the launch of NAPRE, several Indian cities had taken up the Animal Birth Control and Antirabies Vaccination (ABC-ARV) program with the purpose of reducing the number of dogs by employing a humane approach and, in turn, also reducing the number of rabies cases. However, sustained efforts have been lacking due to various reasons. On the other hand, to suit the regional conditions, individual states of India are allowed to develop their own action plan, i.e., state action plan for rabies elimination (SAPRE). Strategies include dog population survey and management and MDV. Further, the effectiveness of the antirabies vaccination programs also needs to be evaluated through monitoring of antibodies elicited following vaccination.

The metropolitan cities of India, including Bengaluru, have a high population of stray dogs. To control the population of these free-ranging dogs, catch–neuter–vaccinate–release (CNVR) was followed by several nongovernmental organizations (NGOs) as part of the ABC-ARV program in Bengaluru to reduce the street dog menace and reduce the cases of dog bites. In September 2019, the Bengaluru City Municipal Corporation [Bruhat Bengaluru Mahanagara Palike (BBMP)] conducted a systematic survey of the stray dog population within its jurisdiction with the support from Veterinary College, Bengaluru, Worldwide Veterinary Services (WVS) India, Mission Rabies, and NGOs by using a web application for collection of data on free-ranging dogs developed by WVS. Accordingly, an estimated 310,000 free-roaming dogs exist in Bengaluru city [8]. Based on this survey, in September 2020, the BBMP initiated mass dog vaccination (MDV) in Rajarajeshwari Nagar (1 of the 8 zones of BBMP) as a pilot study. Thereafter, the MDV program was expanded to all the zones from the year 2021, in addition to the ABC-ARV program with the support of the Mission Rabies team to control rabies in stray dogs.

To attain an epizootiological baseline of herd immunity in a population, 70% of the dog population should be successfully vaccinated against rabies [3]. To know the effectiveness of the MDV program initiated by BBMP, we screened blood samples from free-ranging dogs from all the eight zones of BBMP and studied the role of immune response in rabies PrEP using a rapid fluorescent focus inhibition test (RFFIT) and compared it with an in-house indirect enzyme-linked immunosorbent assay (iELISA). Interferon-gamma (IFN—γ) assay was also employed to evaluate cell-mediated immune (CMI) response in vaccinated dogs. The purpose of this study was to estimate the percentage of dogs with an adequate immune response (≥0.5 IU/mL) to confer protection against rabies through vaccination in a particular zone of BBMP to check the efficacy of MDV.

## 2. Materials and Methods

### 2.1. Study Area

The blood samples from the free-ranging dog population were collected from all the 8 zones of the Bengaluru City Municipal Corporation (BBMP) (Figure 1).

### 2.2. Samples

#### 2.2.1. Approval by Institutional Animal Ethics Committee

The sampling was initiated following the approval (No. VCH/IAEC/2021/27) from the Institutional Animal Ethics Committee (IAEC) of Veterinary College, KVAFSU, Hebbal, Bengaluru.

#### 2.2.2. Sampling

Selection of the total sample size was based on the Cochran formula [9], assuming that 25% of all vaccinated animals were protected. Only 25% was used due to limitations in manpower, budget for the IFN-gamma assay kit, and logistical constraints, but a 95% confidence interval was applied. Accordingly, a sample size of 288 was estimated.

The Cochran formula is
$$n_{0}\;=\;\frac{Z^{2}pq}{e^{2}}$$
wheree is the desired level of precision (i.e., margin of error);p is the (estimated) proportion of the population which has the attribute in question;q is 1 − p;the Z value is found in the Z table.


In the current study, we assumed a protection rate of 25%, so p = 0.25. At 95% confidence and at least 5% plus or minus precision (e = 0.05), a Z value of 1.96 is obtained. Therefore, we have
(1.96)^2^ (0.25) (0.75)/(0.05)^2^ = 288

The number of samples to be collected from each ward was then calculated as per the proportion of the total dog population in the corresponding ward. Three wards were identified for sampling from each zone. These wards were identified through a random sampling tool to include one ward among the highly populated wards, one ward with a medium population size, and the third one among the low-populated wards of that zone (Table 1).

The samples were collected with the help of the BBMP team and nongovernmental organizations (NGOs) involved in rabies vaccination in their respective areas. Free-ranging dogs captured by the net method for vaccination as part of a second round of MDV (after one year) were selected for sampling. Whereas, in wards where the time lapse since vaccination was less than a year, only blood sampling was done.

#### 2.2.3. Samples

Blood samples were collected in two aliquots, one for serum separation and another (3–4 mL) with EDTA. The serum samples were stored at −20 °C for further analysis.

Based on the statistical data, 288 samples were collected from three vaccinated wards of each of the eight zones of the Bengaluru Municipal Corporation. During our sampling, MDV was not implemented in only two wards, although ABC-ARV has been ongoing in these wards since 2007. Hence, in addition to 288 samples, 20 samples were collected from these two wards (10 samples from each ward) to assess their immune status. However, 240 samples were available for analysis as 48 samples were insufficient/unfit (Table 1). In total, 260 samples were analyzed from 26 wards (24 where MDV was implemented and 2 wards where MDV was not implemented but ABC-ARV was ongoing) of eight BBMP zones.

### 2.3. Cells and Virus

BHK-21 cells and challenge virus standard (CVS-11) strain of the rabies virus were utilized for the RFFIT. BHK-21 cells available at KVAFSU-CVA Rabies Diagnostic Laboratory were propagated in Dulbecco’s Modified Eagle’s Medium (DMEM) with 10% fetal bovine serum (FBS) and antibiotics. The titrated CVS-11 strain of rabies virus (RABV) used for RFFIT was obtained from the National Institute of Mental Health and Neurosciences (NIMHANS), Bengaluru.

### 2.4. Rapid Fluorescent Focus Inhibition Test

For RFFIT, a previously standardized protocol [10] was used. The sera were heated to 56 °C for 30 min to inactivate complements [11,12] and serially diluted in a flat-bottomed 96-well microtitre plate (Nunc MaxiSorp™ flat-bottom, Thermo Fisher Scientific, Waltham, MA, USA). Following this, 100 TCID50 of rabies virus was added to all the wells except cell controls and incubated at 37 °C for 90 min. Then, 25,000–30,000 BHK-21 cells/well were seeded to all the wells and incubated at 37 °C for 48 h in a 5% CO_2_ incubator. The media was decanted from the plate without disturbing the monolayer, and the cells were fixed with 70% chilled acetone for 30 min at −20 °C. The fixed cells were incubated with a 1:5 diluted fluorescein-labeled anti-RABV nucleoprotein antibody (Fujirebio Diagnostics, Devault, Malvern, PA, USA) for 1 h at 37 °C. The plates were examined using a fluorescent microscope. The titer of antirabies virus neutralizing antibodies (VNA) was estimated in comparison with WHO reference serum. The highest dilution of the serum, which showed complete neutralization of RABV, was considered the VNA titer. A VNA titer of 0.5 IU/mL serum was considered to reflect adequate immune responses [3,5]. The titer was estimated based on the following equation.
$$\mathrm{Titre}\;(\mathrm{IU}/\mathrm{mL})\;=\;\frac{\mathrm{Highest}\;\mathrm{dilution}\;\mathrm{of}\;\mathrm{test}\;\mathrm{serum}\;\mathrm{showing}\;\mathrm{complete}\;\mathrm{neutralization}\;\times \;\mathrm{Unitage}\;\mathrm{of}\;\mathrm{reference}\;\mathrm{serum}}{\mathrm{Highest}\;\mathrm{dilution}\;\mathrm{of}\;\mathrm{reference}\;\mathrm{serum}\;\mathrm{shows}\;\mathrm{complete}\;\mathrm{neutralization}}$$


### 2.5. Indirect Enzyme-Linked Immunosorbent Assay

The in-house iELISA had been developed using a baculovirus-expressed RABV glycoprotein [13]. Antigen at 500 ng/100 μL/well was coated and incubated at 4 °C overnight. The contents of the wells were discarded, and the plates were washed two times with phosphate-buffered saline (PBS), pH 7.2. Blocking was carried out using 3% bovine serum albumin in PBS along with 0.05% Tween 20 (PBST) at 37 °C for 2 h. After washing three times with PBS, sera diluted in 1% BSA in PBST were added, and the plates were incubated at 37 °C for 60 min. The controls included serially diluted positive and negative sera, conjugate controls, and blank wells. After incubation, the plates were washed two times with PBST, followed by the addition of 100 μL/well of 1:20,000 diluted rabbit anticanine IgG HRP conjugate (Sigma-Aldrich, St. Louis, MO, USA). Following incubation at 37 °C for 60 min and three washes, chromogen-substrate (OPD-H_2_O_2_), and finally, 2.5 N HCl was added to stop the reaction. The optical density (OD) was measured at 492 nm in an ELISA reader.

The OD values obtained for each sample were converted to percent positivity (PP) values. The PP for each sample reading was calculated using the following formula:
(1)$$\mathrm{PP}\;=\;\frac{\mathrm{OD}\;\mathrm{of}\;\mathrm{the}\;\mathrm{test}\;\mathrm{serum}\;\times \;100}{\mathrm{OD}\;\mathrm{of}\;\mathrm{the}\;\mathrm{positive}\;\mathrm{control}}$$


In the in-house iELISA, a cut-off value of 57.1% (PP value) was selected based on work carried out earlier in this laboratory [14]. In brief, the average of all the control PP values were plotted against corresponding neutralizing antibody titers and made into a graph that gave a hyperbola. As per this graph, 0.5 IU/mL of RFFIT corresponded to the 57.1 PP value of iELISA, and hence this value was taken as the cut-off value to determine an adequate immune response in iELISA.

The sensitivity and specificity of the in-house iELISA were determined in comparison with the RFFIT using the statistical formula as per Thrusfield [15].

### 2.6. Interferon-Gamma Enzyme-Linked Immunosorbent Assay

The blood samples (1 mL) were added to 24 well tissue culture plates and stimulated using 5 μL of inactivated rabies antigen (Nobivac-R available at KVAFSU-CVA Rabies Diagnostic Laboratory). The plates were incubated in a humidifier chamber at 37 °C with 5% CO_2_ for 24 h. The supernatant was collected and stored at −20 °C for further analysis.

Sandwich ELISA was performed using the Canine IFN-γ ELISA^BASIC^ kit (Mabtech, USA). The capture antibody (mAb MT13) diluted to 2 μg/mL in PBS (pH 7.4) was added (100 μL/well) to a 96-well flat bottom Nunc Maxisorp plate and incubated overnight at 4–8 °C. The plate was emptied, and 200 μL/well of blocking buffer (0.1% BSA in PBST) was added. The plate was incubated for 1 h at room temperature (RT) and then washed five times with PBST. The samples and standards diluted in blocking buffer were added at 100 μL/well. Assay background control, i.e., wells without standards, were included, and the plate was incubated for 2 h at RT. After washing five times with wash buffer, the detection antibody (mAb MT166-biotin) diluted to 0.5 μg/mL in blocking buffer was added at 100 μL/well and further incubated for 1 h at RT. The plate was washed five times, and streptavidin-HRP diluted 1:1000 in blocking buffer was added at 100 μL/well. The plate was incubated for 1 h at RT, washed, and TMB substrate (product code: 3652-F10) was added at 100 μL/well. The reaction was stopped with 0.2 M H_2_SO_4_, and the optical density was measured at 450 nm in an ELISA reader.

### 2.7. Statistical Analysis

The performance of iELISA was evaluated using Spearman’s rank correlation analysis by comparing it with RFFIT. The strength of iELISA with respect to RFFIT was evaluated using Cohen’s kappa statistics. All the statistical analysis was performed with the help of R studio (Version 1.4.1103) using R base pack 4.0.4 with packages ggplot2.

## 3. Results

The effectiveness of the ongoing MDV program in the free-ranging dog population in Bengaluru was evaluated by assessing the immune responses to rabies vaccination. Humoral immune response was studied by assessing the antibody titer using RFFIT and the in-house iELISA. The cell-mediated immune response was studied using a commercially available IFN-**γ** assay kit.

As evidenced by RFFIT, 171/240 (71%) dogs showed an adequate immune response (≥0.5 IU/mL), whereas only 13/20 (65%) dogs from wards where MDV was not implemented did so (Table 2). The adequacy of response was at least 70% in different zones of BBMP except in Bengaluru South, Dasarahalli, and RR Nagar. Further, the adequate responses were highest at 86.4% (66.7–95.3) in Bengaluru West and least at 53.7% (38.7–67.9) in the RR Nagar zone (Figure 2).

In the in-house iELISA, the cut-off value of 57.1% (PP value) was selected based on work carried out earlier in this laboratory [14]. Accordingly, 87% (209/240) of the samples (Table 2) from the eight zones of BBMP revealed an adequate immune response (≥57.09 PP) in vaccinated wards, and 90% (18/20) of the samples from the two wards where MDV was not implemented also showed adequate antibody titers. The proportion of dogs showing adequate titers was highest at 100% (85.1–100) in Bengaluru West and least at 75.6% (60.7–86.2) in the Rajarajeshwari Nagar zone (Figure 2).

The performance of the in-house iELISA was further evaluated by comparing the results of iELISA with that of the RFFIT. The sensitivity and specificity of in-house iELISA were found to be 100% and 63.3%, respectively (Table 3). The Spearman’s rank correlation (rho value = 0.463 and *p* value < 0.05) indicated the existence of some correlation between the tests, and a kappa value of 0.55 suggested a moderate agreement between RFFIT and iELISA (Figure 3).

The IFN-γ ELISA was employed to assess the cell-mediated immune (CMI) response. The normal range of IFN-γ in dogs is 3 ± 1 pg/mL [16]. In the present study, the samples having IFN-γ concentration above the normal range were considered to have adequate CMI response. Of the 240 dogs from vaccinated wards, 50% (120/240) from all eight zones of BBMP (Table 4) showed adequate CMI response. Interestingly, in the wards where only ABC-ARV was ongoing, but MDV was not implemented, 90% (18/20) of the samples showed adequate CMI response. The highest, 86.4% (66.7–95.3) CMI response was observed in samples from the Bengaluru West zone (Figure 4), and samples from the Yelahanka zone revealed the least at 14.3% (5.7–31.5) CMI response.

Further, in the samples from vaccinated wards, 38% (26/69) of the dogs with inadequate RFFIT titer (<0.5 IU/mL) revealed adequate CMI response. Moreover, 100% (7/7) of the dogs from wards where MDV was not implemented and which had inadequate RFFIT titer showed adequate CMI response.

## 4. Discussion

Mass dog vaccination of stray dogs in Bengaluru municipality was initiated in 2021 and was planned to gradually cover all the free-ranging dogs in all 198 wards of the eight zones. To evaluate the effectiveness of vaccination, sera and blood were collected to evaluate humoral and CMI responses.

In RFFIT, 71% of dogs in vaccinated wards revealed an adequate immune response. This suggests that the citywide MDV program has been effective. Interestingly, 65% of the sera samples from dogs of wards where MDV was not implemented also revealed an adequate immune response. This may be attributed to previous vaccinations (earlier to initiation of MDV) as part of ABC-ARV programs, as protective immunity against rabies following vaccination is thought to last at least 3 years or even more [17]. In these two wards, the ABC-ARV program has been extensively implemented since 2007. Similarly, a study in Sri Lanka reported 60% of free-ranging dogs carried protective antibody titer by RFFIT after one dose of vaccination. In the study, they vaccinated stray dogs and owned dogs and tested serum samples for the presence of antirabies antibodies on days 0, 30, 180, and 360 by employing RFFIT. Among the stray dogs, the percentage of dogs with adequate antibody titer was 1.6 to 6.4 on day 0. However, after one dose of vaccination, an adequate immune response was observed in >70% of the dogs until day 180, but by day 360, the percentage of animals showing adequate titer declined to 43–60%. They concluded that a single dose of antirabies vaccine is not sufficient for the maintenance of antibody titers for a period of one year [18]. In our previous study, where samples were collected from north and south parts of Bengaluru in 2018, 50% of free-ranging dogs had adequate antibodies with the titers ranging from 0.5 to 32 IU/mL in North and 0.5–4 IU/mL in South Bengaluru [19]. However, this study was conducted much before the initiation of the MDV program.

In the current study, 86.4% of the free-ranging dogs in Bengaluru West showed adequate antibody levels, and this may be attributed to the effective vaccination coverage as part of MDV (Figure 5), along with the sustained ABC-ARV program going on in this zone. This suggests that increasing the vaccine coverage helps better seroconversion in the dog population.

However, we observed variations in seroconversion in dogs across the zones. This may be attributed to the many factors which influence the humoral response in dogs, such as age at vaccination, sex, size and breed of dog, nutritional status of the animal, vaccine type, number of vaccinations, time interval from vaccination to sample collection, etc. Studies conducted to evaluate the factors associated with dog rabies immunization have reported that serum antibody levels were associated with vaccine age, genetics of the dog population, number of vaccinations, manufacturer, time interval after most recent vaccination, and sample collection [20,21]. A study in Indonesia investigated both owned and free-roaming dogs for their immune response after rabies vaccination and the factors associated with the development of antibodies within 30 days of vaccination. They observed that the history of vaccination and a good body condition score of the dog were significantly associated with the presence of rabies antibodies at day 0 and the development of antibodies at day 30 [22]. In the current study, a common vaccine was used in the MDV program, and therefore, the type and efficacy of the vaccine could not be a major cause for the variation in the humoral response that we observed, as there is a good response to vaccination by the dogs in certain zones.

The WOAH recommends the use of ELISA for monitoring immune status postvaccination in the framework of rabies control, i.e., in large-scale surveys [7]. In the present study, by iELISA, 87% (209/240) of the dogs from vaccinated wards had adequate antibodies deemed to be protective, whereas, in the wards where MDV was not implemented, 90% of the dogs had adequate antibodies. This may be attributed to the consistent ABC-ARV program going on in these two wards since 2007. Interestingly, the presence of adequate immune response was observed in >70% of the samples from each of the zone when tested by iELISA. A similar study in 2022 reported that only 50% of the dogs had adequate antibody titer, with a higher proportion (66%) among dogs in the north zone than in the south zone (40.5%) of Bengaluru. The study attributed the low proportion to a combination of factors, including lack of sustained vaccination as well as vaccination failure. It was concluded that the high proportion of free-ranging dogs would achieve presumed protection status through continuous annual vaccination through the ABC-ARV program, which was initiated earlier [19]. Another study in Chandigarh, India, where MDV was not initiated, revealed that only 1% of the street dogs had the requisite level of antibodies, as tested by ELISA. However, free mass immunization drives have been conducted in Chandigarh since 2004. Hence, the authors wanted to assess the effectiveness of rabies programs, and thus, both stray dogs and owned dogs were sampled and assessed by ELISA. Surprisingly, adequate antirabies antibody titer was found only in 1% of street dogs and 16% of pet dogs [23]. In a study from Gujarat, India, the proportion was 1.9%, as tested by ELISA [24].

Virus neutralization tests such as RFFIT are tedious and complicated to perform, making them unsuitable for large-scale seroepidemiologic surveillance studies. To overcome this limitation, several ELISA protocols have been developed for detecting antibodies to the rabies virus using monoclonal antibodies [25]. The ELISA has several advantages such as safety (handling of a live virus is not required), simple laboratory requirement (does not require cell culture facility or fluorescent microscope), user friendliness, rapidity, and less technical demand. Poor quality sera can cause cytotoxicity in VNT, which could lead to false positive results. For such samples, indirect ELISA has been shown to be as sensitive and specific as VNT; it also does not require high containment facilities and yields rapid results [26]. Even though ELISA is not directly comparable to neutralization assay results, detection and titration of antiglycoprotein antibodies may be a method of choice on par with RFFIT owing to the above-mentioned advantages. The VN and ELISAs are the recommended tests for monitoring immune status in individual animals or in a population postvaccination in the framework of rabies control. For the purposes of measuring antibody responses to vaccination prior to international animal movement or trade, only VN methods (FAVN test and RFFIT) are acceptable [7]. In the present study, there was a proportionate increase in the immune response by iELISA as compared with RFFIT. The zone-wise assessment of samples indicated the highest responses in Bengaluru West by both RFFIT and IELISA (86.4% and 100%, respectively) and the least response in the bordering RR Nagar zone (53.7% and 75.6%, respectively). The higher percentage observed in iELISA compared to that of RFFIT may be attributed to the ability of iELISA to detect even non-neutralizing antibodies along with neutralizing antibodies, whereas the RFFIT detects only neutralizing antibodies. The sensitivity and specificity of iELISA were found to be 100% and 63.3%, respectively. A Kappa value of 0.55 suggested a moderate agreement between the RFFIT and iELISA.

Interferon-γ is an important cytokine involved in CMI response and is critical for macrophage activation, T-cell proliferation and differentiation, and upregulation of proteins involved in antigen processing and presentation. It exerts an antiviral effect by promoting the lysis and clearance of virus-infected cells and by inhibiting viral gene expression and replication. The CMI response in rabies infection may be an important mechanism in protecting animals against rabies [27,28], but it has not been well investigated. It is important to study the role of cellular response in dogs when antirabies virus neutralizing antibody (RVNA) titers are lesser than 0.5 IU/mL. In a study in humans, a positive correlation was observed between the numbers of IFN-γ producing rabies virus-specific T-cells and RVNA titers, and it was inferred that IFN-γ was important in protection against rabies [29]. In the present study, the IFN-γ levels were above the normal range in 50% and 90% of the samples from dogs of wards where MDV was implemented and not implemented, respectively, suggesting that requisite CMI responses were elicited in a large proportion of dogs. Further, 38% of dogs in vaccinated wards with inadequate RFFIT titer (<0.5 IU/mL) showed adequate levels of IFN-γ (≥3 ± 1 pg/mL), and surprisingly, 100% of the dogs from wards where MDV was not implemented and also had inadequate RFFIT titer (<0.5 IU/mL) showed adequate levels of IFN-γ. This may be attributed to the previous vaccinations undertaken as part of the ABC-ARV program and the T-cell memory [30,31,32].

## 5. Conclusions

The mass vaccination of dogs against rabies is the most rational strategy for interrupting the natural transmission of rabies. The present study was conducted to assess the effectiveness of the ongoing MDV program in the Bangalore City Municipal Corporation (BBMP). The results revealed that 71% of 260 dogs from the 8 zones of BBMP had an adequate immune response for rabies after 51% of the dog population had been vaccinated. The progress can thus be considered good considering the recommendations of WHO, where achievement of at least 70% coverage would be a prerequisite for the prevention or elimination of rabies [3]. This was a preliminary study conducted with a limited number of samples from selected wards of each zone of BBMP. Additional large-scale surveys on previously vaccinated free-ranging dog populations are required in all areas. Our study suggests that along with sustaining the MDV program, the application of simpler tests, such as the in-house iELISA, would pave the way for better implementation of rabies control and elimination programs. Similar drives need to be carried out in other cities in India to realize the goal of eliminating dog-mediated rabies by 2030. Further studies are also required to assess the duration of immunity conferred by rabies vaccines; particularly, the role of cell-mediated immunity needs to be further explored.

## Figures and Tables

**Figure 1 vaccines-11-00888-f001:**
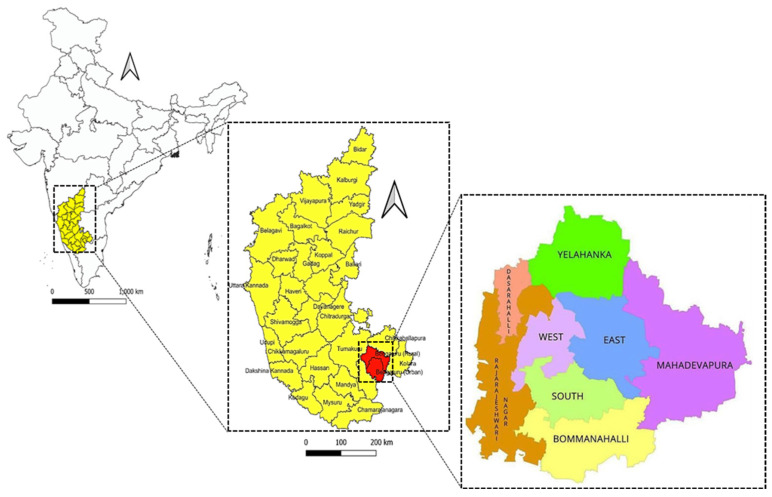
Map of Bengaluru City Municipal Corporation (BBMP) depicting all eight zones.

**Figure 2 vaccines-11-00888-f002:**
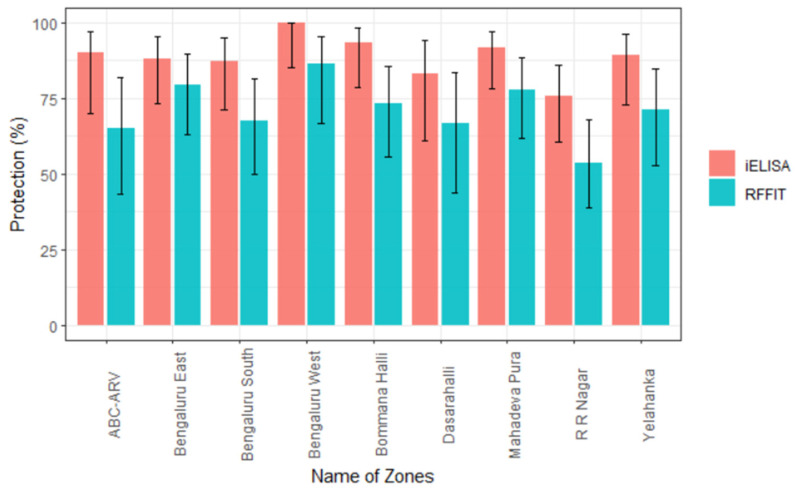
Zone-wise proportion of adequate immune response observed by RFFIT and iELISA (error bars represent range of 95% CI). The percentage of dogs showing adequate immune response is represented as protection (%) on *Y*-axis. The different zones of BBMP area are given on the *X*-axis. ABC-ARV on *X*-axis refers to wards with only ABC-ARV ongoing, but MDV was not implemented.

**Figure 3 vaccines-11-00888-f003:**
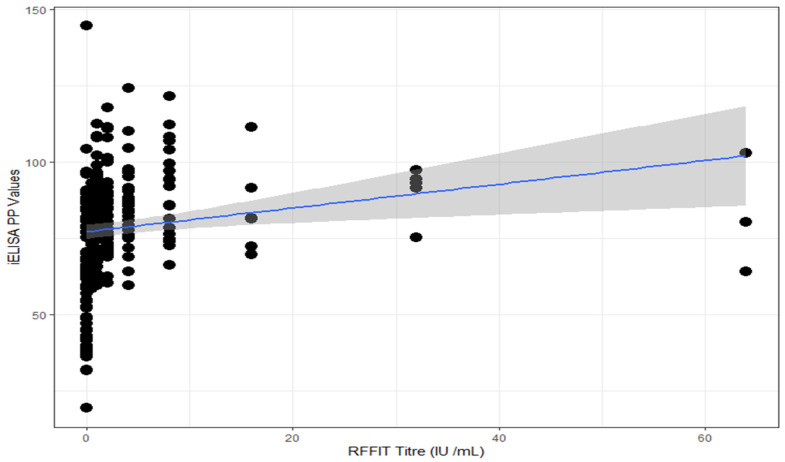
Scatter plot depicting correlation between RFFIT and iELISA. The shaded area represents the confidence interval at 95% level. The blue line is the regression line for the correlation analysis.

**Figure 4 vaccines-11-00888-f004:**
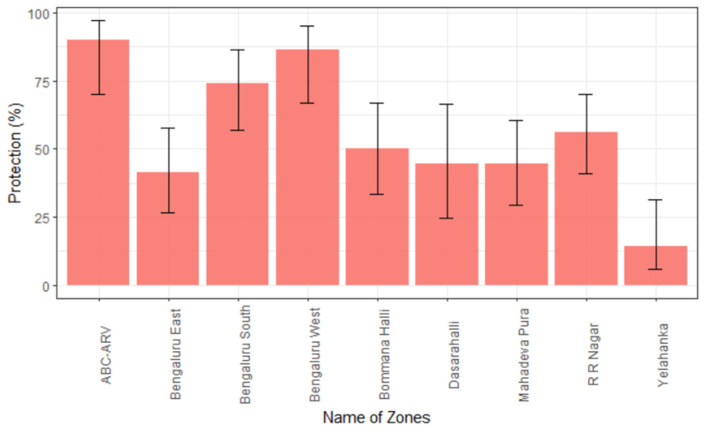
Zone-wise proportion of adequate CMI response in dogs (error bars represent range of 95% CI). The percentage of dogs showing adequate cell-mediated immune response is represented as protection (%) on *Y*-axis. The different zones of BBMP area are given on the *X*-axis. ABC-ARV on *X*-axis refers to wards with only ABC-ARV ongoing, and MDV was not implemented.

**Figure 5 vaccines-11-00888-f005:**
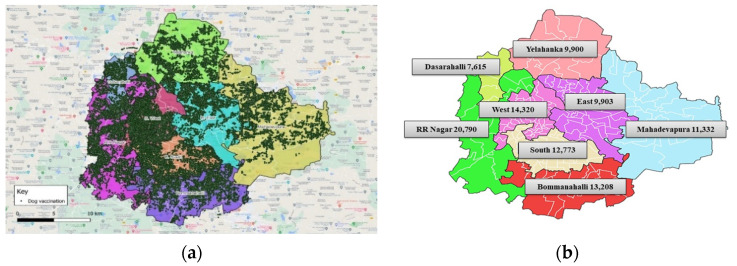
**Number of dogs vaccinated** in various BBMP zones as part of MDV program during 2021. (**a**) BBMP map showing the density of vaccinated dogs in various zones. Each green dot represents one vaccinated dog. (**b**) BBMP map showing the number of vaccinated dogs in each zone.

**Table 1 vaccines-11-00888-t001:** Zone-wise and ward-wise number of dogs sampled for blood collection.

Sl. No.	Name of the Zone	Zone-Wise Estimated Dog Population	% Distribution	Sample Collection from Each Zone	No. of Samples Fit for Analysis
1	Yelahanka	36,217	11.68	34	28
2	RR Nagar	52,968	17.09	49	41
3	Dasarahalli	23,170	7.47	22	18
4	Mahadevpura	46,334	14.95	43	36
5	Bommanahalli	38,940	12.56	36	30
6	B. South	39,562	12.76	37	31
7	B. East	44,303	14.29	41	34
8	B. West	28,481	9.19	26	22
		309,975	99.99	288	240

**Table 2 vaccines-11-00888-t002:** Seroconversion observed in vaccinated dogs.

For 240 Samples	RFFIT		iELISA	
	Adequate Immune Response	Inadequate Immune Response	Adequate Immune Response	Inadequate Immune Response
Number of samples	171	69	209	31
Percentage	71%	29%	87%	13%

**Table 3 vaccines-11-00888-t003:** Two-sided contingency table for sensitivity and specificity of iELISA.

Test		RFFIT		Total	Sensitivity	Specificity	Kappa Value
		Positive	Negative				
**G-protein** **iELISA**	Positive	189	38	227	100%	63.3%	0.55
	Negative	0	33	33			
Total		189	71	260			

**Table 4 vaccines-11-00888-t004:** Cell-mediated immune response observed in vaccinated dogs.

For 240 Samples	Adequate CMIR	Inadequate CMIR
Number of samples	120	120
Percentage	50%	50%

## Data Availability

All the data is presented in the manuscript.

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
