# Peer review of "Assessment of Immune Responses to Rabies Vaccination in Free-Ranging Dogs in Bengaluru, India"

_vaccines, 2023, doi:10.3390/vaccines11050888_

Round 1

Reviewer 1 Report

I thinK that the manuscripto can be improved according to same givem suggestions, as well as the figures.

Author Response

The manuscript has been revised in detail including figures as suggested

Reviewer 2 Report

The paper by Vinay Chavan Prakash Rao ‘Assessment of immune responses to rabies vaccination in free-ranging dogs in Bengaluru, India’ describes a study to assess the effectiveness of mass dog vaccination in Bengaluru and investigate and compare humoral responses tested using iELISA and RFFIT and cellular responses using IFN-y ELISA.  Given the drive to eliminate human rabies globally, the study has merit.

The study appears to have all the data for a reasonable publication however several major improvements are required.

1.     The English needs to be improved for legibility and comprehension.  Proof reading by a fluent English speaker is required before re-submission as the current text is not at the standard expected when a paper is submitted for review.

2.     The Methods require revision to clearly explain how the dogs were selected, how sensitivity and specificity calculations were made, etc.

3.     The figures and legends need to be improved and I suggest combining Fig 2 and 3 so that the two tests can be viewed on the same figure. I also suggest combining Tables 1 and 2 for the same reason.

4.     Most of the Discussion text is repetition of the Methods or Results and should be removed.  Further, the Discussion text that is in the appropriate place requires fleshing out as, as it stands, it is a bit thin. 

5.     Finally, and importantly, the authors need to work on paragraph structure. Currently multiple themes are discussed in single paragraphs which makes the Discussion very hard to follow.  As a word of advice, each paragraph should be focused on ONE (and only one) important point or topic, and that important point or topic should be made crystal clear in the topic sentence (the first sentence of the paragraph), supported by the following sentences, and rounded out by a concluding sentence.  Paragraph design is critical for a reader to be able to follow your points.

6.     Given these points, the Discussion needs to be revised to a) only include Discussion relevant text, b) present the key findings up front, d) discuss ALL results in sufficient detail with examples from the literature where relevant in succinct easy to follow paragraphs. 

Specific issues follow with line number given:

43: Rabies does not circulate in two cycles only. Much of rabies across Africa and Asia is in the rural areas and cannot be described as urban.

45: Reservoirs – cats are not reservoirs; whilst bats are reservoirs for different strains of rabies; whilst in the US small wild carnivores are reservoirs/.  This section needs to be described more carefully.

50: Reference needed

60: I am not sure there is evidence for dog population control preventing human rabies. Please provide ref.

61: English needs to be improved

63: English needs to be improved

69: Should this read ‘The strategies for controlling rabies in dogs include….’?

71-2: English needs to be improved

73: Delete ‘significantly’

80: What is 3.1 Lakh?

91-3: These three tests need to be introduced more thoroughly in the Introduction. What are the properties of the tests, strengths and weaknesses and why are they being compared in this study etc.

100: How were dogs captured?

100-102: Not clear how the dogs were selected or what the statistical tool was? This methodology underpins the whole study and describing this part clearly and in detail is essential for the comprehension of the paper.

109: Usually separate section for approvals text.

123: what cells were in the culture? And more detail needed about the source of the inact. rabies antigen.

127: Need to state at the start of this section what the virus and BHK cells were to be used for.

149: Clarify what considered adequate means

186: What does ‘strength’ mean here?

191-200: Most of this text should be in Methods

204: IR. - has this been defined? As this acronym is not used many times in the paper I suggest, for comprehension, not using an acronym. Same for NC.

Fig 2: Confidence intervals on bars needed.  Legend needs more details.

213-4: This text should be in Discussion

Fig 2 and 3 and Table 1 and 2: suggest combining T1 & 2 and Fig 2 and 3 so can compare directly

221: this sentence needs clarification - especially percent positivity, which if it is to be used should be introduced in the Methods.

222: need to describe in Methods how sens. and sp. were calculated; as not clear how dogs were determined to be vaccinated or not in the vaccination zones.  What was the gold standard that the sensitivity / specificity was calculated with?

224: Cant just state is not equal to zero. And stating that there is ‘some’ correlation is not sufficient. Give r value so we can see what level of correlation there is.

Fig 4: Legend is insufficient

233-7: difficult to understand this text

Fig 5: Text is overlapping – needs to be re drafted. Plus add conf.intervals.  The large variation in the zones – authors could consider statistically comparing the outcome in each zone.

Discussion

251-271: All of this text is not appropriate in Discussion – and is a repetition of Intro.

273-6: This is repetition of results. Its key in presenting discussion text to not simply repeat the results. Rather the results need to be discussed. 

280: Reference needed for this statement

281: Which country was this study in?

282-4: were these dogs in an area that had been vaccinated or not? Relevance to paragraph is not clear.  The opening statement of the paragraph concerns RFFIT.  As such the paragraph should remain focused on this subject and not move onto another topic. 

286-8: This text should be in Results.

292: Run space – not good English. 

294: English needs improving and its not clear how the large pop affected results.

297: This statement is either an assumption, and should be presented as such, or it should be backed up with a reference as the assessment of the no. of dogs covered by MDV in each zone is not a part of this study.

298-9: Its not clear why / how this conclusion is made. No information on the central part of the city etc.  Remember the reader has no idea about these places.

302: Do the authors mean ‘inadequate’ in the NC zone?

305-7: Relevance of this statement is not clear. Was this a vaccinated area? 

307-16: Again, this paragraph opens with discussion about the percent of dogs in  vaccinated wards that were adequately vaccinated.  As such the relevance of listing results from previous studies that assessed antibody levels in dogs in unvaccinated zones is not clear.

318-320: As this is the basis of this paper, further discussion of these tests and the comparative results is warranted.

321-22: This text should be in the Results

324-330: This text should be in the Results

333: How are HI and CMI being compared? Is this comparison described in Methods and Results?

Conclusions: Conclusions should definitely refer to comparison of ELISA and RFFIT and IFy as this is a primary objective of the study.

Author Response

The manuscript has been revised as suggested and attached below

Reviewer 3 Report

In this manuscript ID: vaccines-2221563entitled "Assessment of immune responses to rabies vaccination in free-ranging dogs in Bengaluru, India", Vinay Rao and his colleagues` study was conducted to assess the effectiveness of the ongoing mass dog vaccination (MDV) program in Bruhat Bengaluru Mahanagara Palike (BBMP), Bengaluru, India. A total of 260 blood (wholeblood and serum) samples from stray dogs in 26 wards of 8 BBMP zones were collected and tested by Rapid Fluorescent Focus Inhibition Test (RFFIT) as well as in-house quantitative indirect Enzyme Linked Immunosorbent Assay (iELISA) for humoral response followed by Interferon-gamma (IFN–γ) ELISA for cellular response. This study revealed he mass vaccination of dogs against rabies is the most rational strategy for interrupting the natural transmission of rabies. The outcome of the study revealed that, 71 per cent of 260 samples collected from 8 zones of BBMP had adequate immune response for rabies after 51 per cent of the total dog population was covered. The topic is interesting and should have a appeal to the readership. The authors in general presented a balanced view on the topic of interest. I think there are some good points in this paper.

-          15 references may not be enough.

Author Response

The manuscript has been revised in detail by citing additional references

Round 2

Round 3
